# Immunoinformatics Approach for Epitope Mapping of Immunogenic Regions (N, F and H Gene) of Small Ruminant Morbillivirus and Its Comparative Analysis with Standard Vaccinal Strains for Effective Vaccine Development

**DOI:** 10.3390/vaccines10122179

**Published:** 2022-12-19

**Authors:** Muhammad Hasaan Aziz, Muhammad Zubair Shabbir, Muhammad Muddassir Ali, Zian Asif, Muhammad Usman Ijaz

**Affiliations:** 1Institute of Microbiology, University of Veterinary and Animal Sciences, Lahore 54600, Pakistan; 2Institute of Biochemistry and Biotechnology, University of Veterinary and Animal Sciences, Lahore 54600, Pakistan

**Keywords:** small ruminant morbillivirus, nucleocapsid, fusion, hemagglutinin, Nigeria 75/1, Sungri 96

## Abstract

Background: Outbreaks of small ruminant morbillivirus (SRMV) are regularly occurring in Pakistan despite vaccine availability. This study was designed to identify substitutions within the immunogenic structural and functional regions of the nucleocapsid, fusion, and hemagglutinin genes of SRMV and their comparison with vaccinal strains of Nigerian and Indian origin. Methods: Swabs and tissue samples were collected from diseased animals. RT-PCR was used to characterize selected genes encoded by viral RNA. The study’s N, F, and H protein sequences and vaccinal strains were analyzed for B and T cell epitope prediction using ABCpred, Bipred, and IEDB, respectively. Results: Significant substitutions were found on the C terminus of the nucleocapsid, within the fusion motif region of the fusion gene and in the immunoreactive region of the hemagglutinin gene. Conclusion: Our results emphasize the need for the development of effective vaccines that match the existing variants of SRMV strains circulating in Pakistan.

## 1. Introduction:

Small ruminant morbillivirus (SRMV, also known as peste des petits ruminants) is among the most contagious viral diseases affecting wild and domesticated small ruminants. The virus can cause high mortality (90%) or morbidity (100%) in affected flocks [1]. The virus belongs to the Paramyxoviridae family within the Morbillivirus genus [2]. The RNA genome of SRMV is a single-stranded, negative-sense molecule of 15,948 nucleotides in length [3]. Six structural proteins are encoded in this virus’s RNA genome: the nucleocapsid (N), phosphoprotein (P), matrix (M), fusion (F), hemagglutinin (H), and large protein (L) [4]. The RNA encapsulation is achieved with the formation of the ribonucleoprotein complex (RNP) with the aid of the N protein during viral replication, transcription, and assembly [5]. Hemagglutinin (H) and fusion (F) are the glycoproteins of this virus. For the infection of the cell, fusion of the viral membrane with the outer membrane of the host cell is essential, which is achieved by attaching the viral ligand with the host cell receptor. Through the H protein, the virus is attached to the receptors of the host cell and generally regulates viral entry and adsorption, releasing new viruses and determining their pathogenicity [6].

SRMV was first reported in 1991 in Pakistan; however, in 1994, with the aid of PCR, it was confirmed [7]. Currently, the vaccine used against SRMV in Pakistan is Nigeria 75/1, which belongs to lineage II and originated from West African countries. However, contrary to the vaccinal strain, Pakistan’s field-originated isolates belong to lineage IV. Since then, despite using a live attenuated vaccine, SRMV has remained endemic in Pakistan and has emerged as highly contagious [8]. It is still very arduous to determine the level of vaccine failure through serological monitoring, which limits knowledge of the epidemiology of the disease and its control. Therefore, to aid in disease diagnosis and to design strategies for better disease management in the future, it is essential to determine the nature of existing variants in different regions of the country [9]. The estimated mean evolution rate of the whole genome of this virus is 9.953 × 10^−4^ substitutions per site per year [10]. Since SRMV is endemic to Pakistan, it is imperative to investigate the genetic makeup of field-circulating strains of SRMV for necessary interventions over time. There is a lack of data on the prediction of epitope mapping of selected structural proteins to better elucidate substitutions among vaccine and field-circulating strains. Since the substitution rate is the critical parameter [11] to understand the virus evolution [10], this study aimed to identify amino acid substitutions in the immunogenic regions of the N, F, and H proteins of circulating SRMV strains in Pakistan compared to those of the standard vaccinal strains (Nigeria 75/1 and Sungri 96).

## 2. Materials and Methods

The N, F, and H genes used in this study for comparative analysis with the vaccine strains represented the districts of Lahore, Faisalabad, and Layyah of the Punjab province. The chosen districts had traditional agro-livestock mixed production systems in Pakistan.

### 2.1. Sample Collection and Preparation

Swabs (nasal, oral, and rectal swabs) and other samples (nasal, lacrimal discharge, mucosal, erosion, and blood) from sheep and goats were collected and processed previously by our laboratory for N-gene-based identification and complete genome sequencing [8,10]. The clinical samples represented animals that had clinical symptoms such as high fever, erosion in the nasal mucosa, severe diarrhea, abortions in pregnant animals, and nasal and lacrimal discharges. The number of animals in flocks ranged from 50 to 90, with ages ranging from three months to four years, and each flock had no previous vaccination status. The breeds of sheep in these districts were Thalli, Kajli, and Lohi, while the goat breeds were Daira Din Panah, Beetal, and Teddy.

### 2.2. Retrieval of Protein Sequences and Consensus Sequence Generation

Complete sequences of the N, F, and H genes of SRMV were retrieved from GenBank; KY967608, KY967609, and KY967610 are the accession numbers per the NCBI database. Complete N, F, and H genes for vaccine strains, including Nigeria75/1 (lineage II) and Sungri 96 (lineage IV), were also retrieved in FASTA format. Sequences of the amino acids were aligned through the BioEdit tool ((Hall, North Carolina, United States) Multiple sequence alignment of retrieved protein sequences was performed using the ClustalW algorithm in Bio Edit^®^ version 5.0.6 (Hall, North Carolina, United States)). The Geneious software (Biomatters, Auckland, New Zealand) generated a consensus amino acid sequence for the N, F, and H genes of studied isolates.

### 2.3. Prediction of B Cell Peptides and Their Antigenicity

The B cell epitope prediction was obtained by analyzing the whole protein sequences of the N, F, and H genes of studied isolates and vaccinal strains. The protein sequences were analyzed using the ABCpred server [12] to predict B cell peptides by setting 0.51 as the cut-off score, and 16 mer was set for the length of the epitopes [13]. The Bepipred 2.0 tool was used for the confirmation of the B cell epitope. The B cell epitope predicted in both tools was selected for further analysis. The VaxiJen V2.0 server (Drug Design and Bioinformatics Lab, Sofia, Bulgaria) was used to determine the antigenicity of the peptides, with a threshold value of 0.5 [14].

### 2.4. Prediction of T Cell Peptides

The Immune Epitope Database (IEDB) tool was used for the prediction of T cell peptides, which provides enumerated data on Major Histocompatibility Complex (MHC) ligand elution and MHC binding, as well as experimentally characterized T cell peptides [15].

### 2.5. Prediction of MHC Class I and II Binding Peptides

The whole protein sequences of the N, F, and H genes of studied isolates and vaccinal strains (Nigeria 75/1 and Sungri 96) were analyzed to identify MHC class I and II binding peptides using the Tepi Tool. For the prediction of MHC class I peptides, six mouse alleles H2 (H-2-Kb, H-2-Db, H-2-Kd, H-2-Dd, H-2-Kk, H-2-Ld) were used. Three mouse alleles (H2-Iad, H2-IAb, H2-Ied) were used for MHC class II. The length of predicted peptides was set to be 14 mer and 15 mer for MHC class I and II, respectively.

For the prediction of MHC I peptides, the MHC binding percentile rank or binding affinity is the basis for identifying predicted peptides. The consensus of the percentile rank is the basis of selection, which is ≤ 1% [16], and for MHC class II selection, the top 10% was the basis of consensus of the percentile rank [17].

### 2.6. Conservancy, Immunogenicity, Allergenicity, and Toxicity of Peptides

The degree of conservation of the immunogenicity of peptides in the studied isolates and vaccinal strains was determined using IEDB at the default threshold. The toxicity and allergenicity of peptides were assessed using ToxinPred (Raghava’s group, New Delhi, India.) and the Allertop v 2.0 server (Drug Design and Bioinformatics Lab, Sofia, Bulgaria) at the default threshold, respectively.

### 2.7. Comparative Analysis

Substitutions were identified in the structural and functional motifs of immunogenic regions of the field-prevailing SRMV strain (lineage IV) compared to vaccine strains (lineage II and IV).

## 3. Results

The amplicon sizes of the N, F, and H genes were 1578, 1641, and 1829 nucleotides and were used for subsequent analyses. The similarity index of the N, F, and H genes among the isolates of Lahore, Faisalabad, and Layyah was between 97 and 100%.

### 3.1. Prediction of B Cell Peptides

The ABCpred tool and Bepipred 2.0 tool generated B cell peptides of a fixed length of 16 mer from the consensus sequences of studied isolates and retrieved sequences of the N, F, and H genes of Nigeria 75/1 and Sungri 96. In the studied isolates, 49 peptides were generated from the N gene, 42 from the F gene, and 54 from the H gene. Similarly, in the vaccinal strain of Nigeria 75/1, 48 peptides were obtained from the N gene, 51 from the F gene, and 62 from the H gene. In the Sungri 96 strain, 51 peptides from the N gene, 54 from the F gene, and 61 from the H gene were generated. These epitopes were screened for further analysis to determine their antigenicity using the Vaxigen server. Among them, peptides that had antigenicity above 0.5 were utilized for further analysis.

### 3.2. Prediction of T Cell Epitopes

Based on selected alleles against MHC I and II, 52 peptides from the N gene and 66 peptides from the F gene were produced, as well as 128 peptides from the H gene of the studied isolates. Similarly, 47 peptides from the N gene, 67 from the F gene, and 112 from the H gene of the Nigeria 75/1 strain were produced. The genes of Sungri 96 generated 49 peptides from the N gene, 72 from the F gene, and 113 from the H gene. Among them, peptides found to be common among B cells from MHC class I and MHC class II were utilized for the comparative residue analysis.

### 3.3. Conservancy, Immunogenicity, Allergenicity, and Toxicity of Peptides

All the peptides of the studied isolates and vaccinal strains were 100% conserved and immunogenic. No allergenicity or toxicity of peptides was observed.

### 3.4. Comparative Residue Analysis

With reference to vaccine strains, a comparative analysis for SRMV was accomplished by identifying the substitutions in the structural and functional motifs in the immunogenic regions of the field-prevailing SRMV.

#### 3.4.1. Substitution Analysis of Nucleocapsid Protein of Studied Isolates and Vaccinal Strains

Significant substitutions were identified in a peptide of Sr. no. 3, 4, and 7 at the C terminus region, where valine (V) was replaced with alanine (A) at 407, compared with Nigeria 75/1 (Table 1).

#### 3.4.2. Substitution Analysis of Fusion Protein of Studied Isolates and Vaccinal Strains

A substitution was identified within the peptide Sr. no. 4 found within the fusion peptide domain, in which valine (V) was replaced with alanine (A) at 110, compared with Nigeria 75/1 (Table 2).

#### 3.4.3. Substitution Analysis of Hemagglutinin Protein of Studied Isolates and Vaccinal Strains

Notable substitutions were found among the peptides Sr. no. 6, 7, and 8 within their immunodominant regions. In the peptide of Sr. no. 6, tyrosine (Y) was replaced with arginine (R) at aa574 compared with Nigeria 75/1. In the peptide of Sr. no. 7 and 8, leucine (L) was replaced with arginine (R) at aa547, and tyrosine (Y) was replaced with phenylalanine (F) at aa552, respectively, when compared to Sungri 96 (Table 3).

## 4. Discussion

Several outbreaks of SRMV have been reported in Pakistan since its identification [3]. Most of them were based on their clinical history and either on antibody or antigen detection. However, the molecular characterization of SRMV is essential in understanding its genetics.

Rapid changes in the genomes of RNA viruses are considered a complex mechanism that creates different variants during the replication of the RNA genome [18]. Since low pathogenicity and adaptability can be achieved by limitations in genetic changes within the viral population [19], the substitution rate is a key parameter in understanding the evolution of the virus [18]. The mean evolutionary rate of the N, F, and H proteins is estimated at 1.1 × 10^−3^, 2.01 × 10^−3^, and 1.47 × 10^−3^, respectively [10]. Despite a single serotype and minor antigenic divergence, the genome plasticity of SRMV might elucidate its potential to emerge and adjust in different geological regions and hosts [10]. Therefore, our study aimed to identify substitutions in these proteins by comparing them with standard vaccinal strains by in silico analysis.

The advancements in the bioinformatics field have had an extraordinary impact on the field of immunology through the development of novel vaccines via computational biology [20]. B and T cell multi-peptide-based vaccines have been designed by researchers from all over the globe using these computational methods, against several viral diseases, such as HIV [21], influenza [22], and dengue fever [23]. Therefore, the current study was developed to identify B cell and T cell peptides of the N, F, and H genes of vaccinal strains and field strains of SRMV and identify the substitutions between them using various bioinformatics tools. All predicted B and T cell peptides indicated a desirable assessment of their immunogenicity, antigenicity, conservancy, and surface accessibility, as suitable criteria for vaccine development, by using different computational algorithms along with defined cut-off or threshold values [24].

Peptide (^328^AYPLLWSYAMGVGVE^342^) is within the RNA binding motif found conserved in the nucleocapsid gene of SRMV. Previously reported results indicate amino acid conservancies at this position in the nucleocapsid gene [25]. However, the mutation was found in the C terminal region of the studied isolate strain in (^406^TVRGTGPRQAQVSF^419^) at the position of 407 (V407A), where valine (V) is replaced with alanine (A), by comparing it with Nigeria 75/1. Valine (V) promotes the formation of infectious particles [26], whereas alanine is physicochemically innocuous and constitutes a deletion of the side chain at the β-carbon [27]. However, a mutation within the C terminus portion of the nucleocapsid protein may have an effect on the enhancement of the persistency of viral infection for a long time, as this is considered a critical region for the replication of the virus [5].

Significant motifs are found in fusion proteins such as cleavage sites [28], leucine zipper, and signal peptide structures. However, the mutation was identified within the fusion peptide (^99^TLTPGRRTRRFAGAV^113^) at the position of 110 (V110A) by comparing it with Nigeria 75/1. Moreover, cleavage sites (^103^RRTRR^108^) remain conserved [5]. The adaptation of the virus to the environment, and its signaling, is highly dependent upon the signal peptide region, the cleavage site for the virulence of the virus. The glycine residue is essential for the activity of fusion of the membrane [29]; its fusion to the host cells can be affected by the substitution of this amino acid [30].

The highest numbers of substitutions were found in the hemagglutinin gene compared with other genes. These results are consistent with previously reported observations where greater hypervariability and high divergence have been revealed in the hemagglutinin gene of SRMV [31]. It is said that in the H protein, immuno-dominant regions are present within B cell peptides using monoclonal antibodies, which indicates that regions from 538 to 609 amino acids are immunoreactive, especially aa574 [32]. There may be an important position of amino acid 574 in the peptide of the H protein. The disadvantage of SRMV vaccine protection may be attributed to the changes in the amino acid at positive selection sites [33]. Our study on the H gene indicated a substitution in the peptide (^570^CFPWRHKVWCYHDC^583^) at 574, where tyrosine (Y) was replaced with arginine (R) (Y574R) in our studied isolate, which was observed by comparing it with the Nigeria 75/1 strain. The tyrosine (Y) phosphorylation in the intracellular transportation of M1 controls the process of viral replication [34]. In contrast, arginine is essential for late viral functions, such as developing viral coat proteins and producing infectious virions [35].

## 5. Conclusions

A substitution was found at the significant sites of the N, F, and H proteins, which are considered essential for functional and structural integrity. There are limited data available about the genome of SRMV, even though it is endemic and outbreaks are still occurring in Pakistan. The genomic analysis performed in this study does not indicate that the current vaccine cannot neutralize the generated escape mutants. To determine the neutralization of the present vaccine against SRMV, experiments need to be performed to help control and eradicate SRMV.

## Figures and Tables

**Table 1 vaccines-10-02179-t001:** Comparative substitution analysis of nucleocapsid protein of studied isolates and vaccinal strains (Nigeria 75/1 and Sungri 96).

Sr. No.	Strain	Peptide	Substitution
Residue Analysis of Peptides Found in B Cells and MHC Class I			
1	Nigeria 75/1	^355^SYFDPAYFRLGQEM^368^	-
	Study isolate		
	Sungri 96		
2	Nigeria 75/1	^328^AYPLLWSYAMGVGV^341^	-
	Study isolate		
	Sungri 96		
3	Nigeria 75/1	^406^T**V**RGTGPRQAQVSF^419^	V 407 A
	Study isolate	T**A**RGTGPRQAQVSF	
	Sungri 96	TARGTGPRQAQVSF	
Residue Analysis of Peptides Found in B Cells and MHC Class II			
4	Nigeria 75/1	^404^ERT**V**RGTGPRQAQVS^418^	V 407 A
	Study isolate	ERT**A**RGTGPRQAQVS	
	Sungri 96	ERTARGTGPRQAQVS	
**5**	Nigeria 75/1	^328^AYPLLWSYAMGVGVE^342^	G 342 E
	Study isolate	AYPLLWSYAMGVGV**E**	
	Sungri 96	AYPLLWSYAMGVGV**G**	
Residue Analysis of Peptides Found in B Cells, MHC Class I, and MHC Class II			
6	Nigeria 75/1	^328^AYPLLWSYAMGVGVEL^343^	
	Study isolate	AYPLLWSYAMGVGV**E**L	G 342 E
	Sungri 96	AYPLLWSYAMGVGV**G**L	
7	Nigeria 75/1	^404^ERT**V**RGTGPRQAQVSF^419^	V 407 A
	Study isolate	ERT**A**RGTGPRQAQVSF	
	Sungri 96	ERTARGTGPRQAQVSF	

**Table 2 vaccines-10-02179-t002:** Comparative mutational analysis of fusion protein of studied isolates and vaccinal strains (Nigeria 75/1 and Sungri 96).

Sr. No.	Strain	Peptide	Substitution
Residue Analysis of Peptides Found in B Cells and MHC Class I			
1	Nigeria 75/1	^155^QAIEEIRLANKETI^168^	-
	Study isolate		
	Sungri 96		
2	Nigeria 75/1	^264^RVTYVDTRDYFIIL^277^	-
	Study isolate		
	Sungri 96		
3	Nigeria 75/1	^431^REYPDSVYLH**E**IDL^444^	
	Study isolate	REYPDSVYLH**K**IDL	E441 K
	Sungri 96	REYPDSVYLHKIDL	
Residue Analysis of Peptides Found in B Cells and MHC Class II			
4	Nigeria 75/1	^99^TLTPGRRTRRF**V**GAV^113^	V 110 A
	Study isolate	TLTPGRRTRRF**A**GAV	
	Sungri 96	TLTPGRRTRRFAGAV	
5	Nigeria 75/1	^274^FIILSIAYPTLSEIK^288^	-
	Study isolate		
	Sungri 96		
6	Nigeria 75/1	^431^REYPDSVYLH**E**IDLG^445^	E 441 K
	Study isolate	REYPDSVYLH**K**IDLG	
	Sungri 96	REYPDSVYLHKIDLG	
Residue Analysis of Peptides Found in B Cells, MHC Class I, and MHC Class II			
7	Nigeria 75/1	^431^REYPDSVYLH**E**IDLGP^446^	E 441 K
	Study isolate	REYPDSVYLH**K**IDLGP	
	Sungri 96	REYPDSVYLHKIDLGP	

**Table 3 vaccines-10-02179-t003:** Comparative mutational analysis of hemagglutinin protein of studied isolates and vaccinal strains (Nigeria 75/1 and Sungri 96).

Sr. No.	Strain	Peptide	Substitution
Residue Analysis of Peptides Found in B Cells and MHC Class I			
1	Nigeria 75/1	^102^EVGIRIPQKFSDLV^115^	-
	Study isolate		
	Sungri 96		
2	Nigeria 75/1	^311^**S**GVPKREPLVVVIL ^324^	S311R, K317E
	Study isolate	**R**GVPKR**E**PLVVVIL	
	Sungri 96	SGVPKR**K**PLVVVIL	
3	Nigeria 75/1	^466^MINTIGFP**D**R**A**EVM^479^	D474N, A476T
	Study isolate	MINTIGFP**N**R**T**EVM	
	Sungri 96	MINTIGFPNR**A**EVM	
4	Nigeria 75/1	^539^VYYIYDTGRSSSYF^552^	L547R, Y552F
	Study isolate	VYYIYDTG**R**SSSY**F**	
	Sungri 96	VYYIYDTG**L**SSSY**Y**	
5	Nigeria 75/1	^548^SSSYFYPVRLNFRG^561^	Y552F, R560K
	Study isolate	SSSY**F**YPVRLNF**K**G	
	Sungri 96	SSSY**Y**YPVRLNF**R**G	
6	Nigeria 75/1	^570^CFPW**Y**HKVWCYHDC^583^	Y574R
	Study isolate	CFPW**R**HKVWCYHDC	
	Sungri 96	CFPWRHKVWCYHDC	
Residue Analysis of Peptides Found in B Cells and MHC Class II			
7	Nigeria 75/1	^539^VYYIYDTGRSSSYFY ^553^	L547R, Y552F
	Study isolate	VYYIYDTG**R**SSSY**F**Y	
	Sungri 96	VYYIYDTG**L**SSSY**Y**Y	
Residue Analysis of Peptides Found in B Cells, MHC Class I, and MHC Class II			
8	Nigeria 75/1	^539^VYYIYDTGRSSSYFYP^554^	L547R, Y552F
	Study isolate	VYYIYDTG**R**SSSY**F**YP	
	Sungri 96	VYYIYDTG**L**SSSY**Y**YP	

## Data Availability

Data will be available on demand.

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
