# Peer review of "Immunoinformatics Approach for Epitope Mapping of Immunogenic Regions (N, F and H Gene) of Small Ruminant Morbillivirus and Its Comparative Analysis with Standard Vaccinal Strains for Effective Vaccine Development"

_vaccines, 2022, doi:10.3390/vaccines10122179_

Round 1

Reviewer 1 Report

In this manuscript, authors tried to analyze the B and T cell epitopes on N, F, and H genes of SRMV endemically infected in Pakistan ruminants, and compared them with those on the vaccine strain Nigeria75/1.  The vaccine strain belongs to lineage II genotypes, while the Pakistan endemic virus belong to lineage IV genotype.

Manuscript type should be: brief communication.

line 21-22 suggest change:  "emphasize the need for the development of a more effective"

line 67: indicate what are these "live animals"? goat, sheep, cattle ? and which breeds?  We do not know if this SRMV can infect larger ruminants.

line 103, 105: the use of mouse allele is not appropriate.  Immunologically mouse is quite distant from small ruminants? Have you checked the availability of other animal species, such as bovine, humans etc, in the database?

line 109: the  "Moutaftsi et al, 2006" is not listed in references.

Tables 1-3: replace "study isolate" with "study isolates" if you are having sequences from more than 1 animals.

Tables 1-3: you list the substitution by subcategory "B-cell and MHC-class I" and "B-cell and MHC-class II" and "B-cell, MHC class I, MHC class II" etc.  What is the difference among these 3 subcategories?  Why not  just list one category that mimics the natural immunity generation the most.

Discussion:

The situation in Pakistan is endemic, so you cannot discuss the amino acid substitution from an "acute infection" point of view.  The pressure for selection should be always positive for endemic status.  So to discuss these amino acid substitutions with antibody titers or cell mediated immunity data found in the population should be more significant.

Reviewer 2 Report

The authors present an interesting study on small ruminant morbillivirus (SRMV). They have collected field samples from animals exhibiting clinical signs consistent with SRMV infection. A total of 18 animals were sampled with all testing positive for the virus of interest. The authors have subsequently amplified and sequenced the Nucleocapsid, Fusion, and Hemagglutinin genes for the key morbillivirus proteins. They have subsequently used available bioinformative tools to identify putative epitopes from the nucleocapsid, fusion, and hemagglutinin proteins. Comparisons were subsequently made between these putative epitopes from two vaccine strains of SRMV and the field strains detected in the current study.

Line 12 suggest revision “Outbreaks of”

Line 13 suggest revision “This study was designed to”

Line 17 suggest revision “characterization of selected genes encoded by the viral RNA.”

Line 21 suggest revision “Our results emphasize the need for the development of effective vaccines that match the existing variants of SRMV strains circulating in Pakistan.”

Line 23 please review the keywords, abbreviations and acronyms are not good keywords

Line 27 suggest revision “Small ruminants morbillivirus (SRMV, also known as peste des petis ruminants) is among the most contagious viral disease affecting wild as well as domesticated small ruminants. The virus can cause high mortality (90%) or morbidity (100%) in affected flocks.”

Please check the suggested changes for accuracy.

Line 31 suggest revision “phocine morbillivirus and dolphin morbillivirus.”

Though I think the later should be “cetacean morbillivirus”. Please review.

Line 32 suggest revision “The RNA genome of SRMV is a single started, negative sense molecule of 15,948 nucleotides in length.”

Line 33 suggest replacing “6” with “six”

Line 57-60 suggest revision

“Therefore, the aim of this study was to identify amino acid substitutions in the bioinformatically identified immunogenic regions of the N, F, and H proteins of circulating SRMV strains in Pakistan compared to those from the standard vaccinal strains of Nigeria 75/1 and sungri96.”

Please check the suggestion conveys the intended message.

Also note that throughout the manuscript the authors regularly refer to genes and substitutions, which is incorrect. They should carefully check the manuscript to ensure the correct terms are used. If the authors are referring to changes in either the viral genome or transcripts from the genome, these should be referred to as “mutations”. If these mutations result in a change in the amino acid of a particular protein, this is referred to as a “substitution”.

Line 63 – please provide a more detailed, ideally brief, description of these districts and how that might influence the detection of SRMV.

Line 67 – What type of animals? Please provide a brief description of the sampled populations. What was their vaccination status? Were flocks selected at random or known to have a history of SRMV? Based on the results, 18 animals were sampled, how were these distributed across the three districts?

Line 88 – see previous comment. You cannot align amino acid sequences of a “gene”.

Line 113 suggest revision “The degree of conservation of immunogenicity”

Line 122-127 suggest revision “

Eighteen animals were identified that exhibited clinical signs consistent with SRMV. All were found to be found positive for the virus by successfully amplifying the three most important genes, namely N, F, and H, using RT-PCR. The open reading frames (ORF) of these genes were also amplified by RT-PCR assays, from samples representing Lahore, Faisalabad and Layyah. The amplicons sizes were 1578, 1641, and 1829 nucleotides for the N, F, and H genes, respectively. The sequences of these amplicons were used for subsequent analyses.”

Line 128 suggest revision “genes”

What were the levels of conservation at the amino acid level, ie identity and similarity? The subsequent epitope analyses suggest 100%. Is this correct?

Line 157 – Suggest revision.

It is cumbersome to have the complete peptide sequence in the text and unnecessary given it is shown in Table 1. It would be easier to follow the text if the peptides being discussed were referred to by the “Sr. No” as shown in the table. Thus the peptide sequence shown on lines 157 and 158 would be peptide Sr. no 3 with the amino acid substitution referred to as V407A.

How was this substitution determined to be “significant”?

Line 160 Table 1

The table title states “genes” but amino acid sequences are shown.

Highlighting amino acid substitution with black text on a dark blue background makes them very difficult to see. Please revise.

Does the lack of a motif sequence for “study isolate” and “Sungri96” mean they are identical to Nigeria75/1? IF so this should be stated either in the table title or added as footnote.

Similar comments for the subsequent tables and their descriptions as well.

Line 192 Please provide a reference for these substitution rates.

The discussion mostly paraphrases the results, although there is some effort to link the detected amino acid substitutions with the know functions of the viral proteins.

Less effort has been made to link the detected substitutions with vaccines. For example, on the basis of this study would one vaccine be preferred compared to the other?

The authors should also address the potential weakness of their study in that the prediction tools used may not accurately reflect how antigens are processed by SRMV hosts. For example, section 2.6 predicts MHC class I and class II binding peptides using murine alleles. Would the results of the current study be better (or worse) if host specific predictions were feasible?

Line 234 suggest replacing “restricted” with “limited”

Reviewer 3 Report

The authors discuss using an immunoinformatics approach to develop a vaccine for small ruminants morbillivirus. The currently used live attenuated vaccines for small ruminants morbillivirus are effective against the different genotype of small ruminants morbilivirus. 

The minor differences in epitopes described in the paper does not impact the effectiveness of the currently used PPR vaccines. If the authors would present this evidence, then the paper would have a suitable rational for pursuing the work.

Reviewer 4 Report

There have been numerous outbreaks of SRMV in Pakistan since its initial identification in 1991.  These outbreaks continue to arise despite the use of the Nigeria 75/1 vaccine.  This study sets out to characterize a total of 18 field isolates with respect to amino acid substitutions in B and T cell epitopes in the viral N, F and H protein sequences that may be responsible for their ability to escape the protection afforded by vaccination.  To predict B cell epitopes, a series of 16-mer fixed length peptides, 49 peptides from the N gene, 42 from the F gene and 54 from the H gene, were screened.  A similar approach was used for the identification of T cell epitopes.  Significant substitutions were identified in the C terminus of the N protein, within the fusion peptide of the F protein and in an immunodominant region of the H protein.

This study makes a strong case for continued monitoring of SRMV isolates as they arise with the resulting data factored into the design of vaccine strategies.  One major criticism of the study with respect to the identification of B cell escape mutations concerns those residing in conformational epitopes.  Certainly, a high percentage of antibody binding epitopes will be conformational in nature.  Identification of substitutions in these epitopes would be solely by chance, as the study is apparently not designed to target them.  Some acknowledgement of this limitation in the study should be added to the manuscript.

Round 2

Reviewer 1 Report

I still think to choose "mouse allele" for MHC analysis was not a good idea, because the final application will be on ruminants.  You cannot replace the "goal" with the "means", i.e. the lab animals. A published idea that you adopted may be potential misleading.  You have to at least address this issue as a limitation of the study.

Reviewer 2 Report

The authors have adequately addressed the majority of the comments and suggestions made from my initial review of their manuscript.

However, I remain a little confused by the response to this comment (comments addressed and their corresponding replies have been "crossed" out):

Point 19: Line 160 Table 1

The table title states “genes” but amino acid sequences are shown.

Highlighting amino acid substitution with black text on a dark blue background makes them very difficult to see. Please revise.

Does the lack of a motif sequence for “study isolate” and “Sungri96” mean they are identical to Nigeria75/1? IF so this should be stated either in the table title or added as footnote. 

Similar comments for the subsequent tables and their descriptions as well.

Response 19: Table tile showing gene is replaced to protein. Highlighting aa substitution is revised. Lack of motif sequence does not mean that they are identical. We focused on substitution which lies within the structural and functional motifs of known viral protein

What does the lack of a motif sequence for a strain mean then?

Take for example Sr. No. 1 in Table 1. The peptide sequence seems to align with the study isolate, with no peptide shown for Nigeria75/1 or Sungri96. Does this mean the algorithm did not predict this MHC Class I motif for Nigeria75/1 or Sungri96? If so this should be explained. Or to rephrase the question - why are peptides not shown for each virus strain in the tables?

Reviewer 3 Report

The authors have not demonstrated that the identified mutations are important for the existing live attenuated PPR vaccines to be ineffective.

The authors can demonstrate this by taking sera from vaccinated animals and comparing the virus neutralization activity to the virus used in the vaccine and the field virus.

There is no evidence that these mutations identified in the study play any role in vaccine failure.
